# Facilitated Transport of Copper(II) across Polymer Inclusion Membrane with Triazole Derivatives as Carrier

**DOI:** 10.3390/membranes10090201

**Published:** 2020-08-27

**Authors:** Bernadeta Gajda, Radosław Plackowski, Andrzej Skrzypczak, Mariusz B. Bogacki

**Affiliations:** 1Czestochowa University of Technology, Department of Metallurgy and Metal Technology, 42-200 Czestochowa, ul. Armii Krajowej 19, Poland; 2Poznań University of Technology, Faculty of Chemical Technology, Institute of Chemical Technology and Engineering, 60-965 Poznań, ul. Berdychowo 4, Poland; radoslaw.plackowski@gmail.com (R.P.); Andrzej.Skrzypczak@put.poznan.pl (A.S.)

**Keywords:** polymer inclusion membrane, copper transport, 1-alkyl-1,2,4-triazole, selective transport

## Abstract

This study investigates copper(II) ion transport through a polymer inclusion membrane (PIM) containing 1-alkyl-1,2,4-triazole (n = 8, 9, 10, 11, 12, 14), *o*-nitrophenyl octyl ether as the plasticizer and cellulose triacetate as the polymer matrix. The feeding phase was a solution of 0.1 mol/dm^3^
*CuCl*_2_ and an equimolar (0.1 mol/dm^3^) mixture of copper, nickel, and cobalt chlorides with varying concentrations of chloride anions (from 0.5 to 5.0 mol/dm^3^) established with NaCl. The receiving phase was demineralized water. The flow rate of the source and receiving phases through the membrane module was within the range from 0.5 cm^3^/min to 4.5 cm^3^/min. The tests were carried out at temperatures of 20, 30, 40 and 50 °C. Transport of NaCl through the membrane was excluded for the duration of the test. It was noted that the flow rate through the membrane changes depending on the length of the carbon chain in the alkyl substituent from 16.1 μmol/(m^2^s) to 1.59 μmol/(m^2^s) in the following order: *C*_8_
*> C*_9_
*> C*_10_
*> C*_11_
*> C*_12_
*> C*_14_. The activation energy was 71.3 ± 3.0 kJ/mol, indicating ion transport through the PIM controlled with a chemical reaction. Results for transport in case of the concurrent separation of copper(II), nickel(II), and cobalt(II) indicate a possibility to separate them in a selective manner.

## 1. Introduction

Distinctive properties of copper in terms of electric and thermal conductivity, resistance to corrosion, and malleability make it one of the most widely applied non-ferrous metals [1,2]. Copper metal is especially valuable in electric and electronic industries [3]. Currently, copper is manufactured mainly by pyrometallurgical processes [4,5,6]. A significant flaw of this process is the production of large amounts of sulphur dioxide [7,8,9,10]. Additionally, waste generated in the course of pyrometallurgical processes contain heavy metals that are harmful to the environment [11]. Furthermore, high energy consumption and low level of metal recovery are significant flaws [12].

Increased production and related increase in the demand for various metals result in a decrease in their deposits on Earth. Poorer and poorer sources are being mined, despite their lower metal contents. It is quite probable that the available deposits will prove to be insufficient in the near future [13]. For this reason, recovering of metals from various wastes is attracting increasing attention worldwide. The so-called wet processes will prove to be the best solution to this problem.

For the last half-century, hydrometallurgical processes have had an increasingly important role in the metal industries. Hydrometallurgy and, lately, biohydrometallurgy allow recovery of metals even from very poor deposits [14,15,16]. It is estimated that about 18% of the copper manufactured in the world is recovered by hydrometallurgical processes [17]. The main advantages of applying hydrometallurgical and biohydrometallurgical processes is the relatively low energy consumption, versatility of the method in providing different metals simultaneously, and lack of requirement for any large plants. Because of these characteristics, various poor resources, including polymetallic resources, can be processed [18,19]. The production of electronic equipment causes an increasing amount of various waste to be generated—e.g., when electric and electronic devices are discarded after use. These methods are also more environmentally friendly than pyrometallurgical methods [8,16,20,21,22,23,24,25,26]. From an environmentalist’s point of view, it is also important that hydrometallurgical processes can be adjusted to produce pure sulphur instead of sulphur dioxide [7,27].

A number of research groups worldwide have carried out research on hydrometallurgical methods. Two main directions of research can be distinguished: improving the efficiency of leaching and selective separation of metals from a complex solution after leaching [20,28,29]. Mixed processes combining pyro- and hydrometallurgical processes are also being researched [30].

One of possible solutions for the selective separation of metal ions after leaching is applying membrane processes, in particular polymer inclusion membranes (PIMs). These methods, as opposed to extraction methods, allow for the elimination of large amounts of chemical compounds used in classic extraction [31,32]. It is possible, because when working with PIMs, one uses low amounts of extractants that serve the role of carriers of metal ions [33]. This provides an additional benefit: it enables the use of expensive but more selective extractants.

To transport copper ions through a PIM, one can use various extractants: phosphoroorganic, e.g., di-(2-ethylhexyl)phosphoric acid (D2EHPA) [3]; aliphatic and aromatic oximes, e.g., 2-hydroxy-5-nonylbenzaldehyde oxime (LIX860) [34,35,36]; ethers [37]; quaternary ammonium salts, e.g., trioctylmethylammonium chloride (Aliquat-336) [38]; or amines, e.g., trioctylamine (TOA) [39]. However, the search for new selective extractants/carriers of metal ions continues. In recent years, ionic liquids have been used increasingly in research projects focused on identifying new extractants [40,41,42,43,44,45,46].

Imidazoles [47] and triazoles [48] are the precursors of many currently applied ionic liquids. These compounds are five-membered aromatic compounds containing 2 or 3 atoms of nitrogen. They belong to a large group of heterocyclic aromatic compounds: azoles. In general, azoles are five-membered aromatic amines containing from one to five atoms of nitrogen in their cyclical structure. Apart from an atom of nitrogen, they can also contain an atom of oxygen or sulphur.

Aromatic amines are not as easily protonated as aliphatic amines. Therefore, they can serve as N-donor extractants for such transition group metals as copper, nickel, cobalt, or zinc [49,50]. Alkyl derivatives of imidazoles, entering the hydration sphere of ions—such as copper(II), nickel(II), or cobalt(II)—create complexes with the general form of MCl_2_L_2_, where L is an organic ligand [51,52]. Because of these properties, azoles, imidazoles [51,52,53,54,55,56,57,58,59,60], and triazoles [59,60] in particular, are used as extractants or carriers of ions for selective extraction and separation of not only transition group metals but also organic acids [61,62,63].

Triazoles are widely applied in many fields, from medicine [64,65] to corrosion inhibition [66,67]. Furthermore, owing to their good thermal stability, they are used in fuel cells as one of the ingredients of the electrolyte [64].

Three atoms of nitrogen in a triazole molecule can reside in the positions of: 1, 2, 3 or 1, 2, 4. These compounds, containing three atoms of nitrogen with free pairs of electrons, are capable of creating both hydrogen bonds and coordinate bonds with ions of such metals as cobalt, nickel, copper, zinc, or cadmium [59,68]. Additionally, triazoles are characterized with lower basicity than imidazoles, while their pKa is lower than pKa of imidazoles [68,69].

Imidazole derivatives can be used as selective extractants/carriers of transition group metal ions. 1-octylimidazole-2-aldoxime allows for the effective separation of copper(II) ions from sulphuric(VI) acid solutions containing ions of nickel(II), cobalt(II), cadmium(II), and zinc(II) [70]. Authors state that separation of metal ions occurs in the following order: *Cu*^2^^+^
*> Ni*^2^^+^
*> Zn*^2^^+^
*> Cd*^2^^+^
*> Co*^2^^+^. A derivative of imidazole, i.e., 1-octyl-2-(2′-pyridyl)imidazole is used for the selective extraction of nickel(II) from sulphuric(VI) and chloride acidic solutions containing ions of various metals [71]. In that case, the authors inform on the following order of separating ions: *Ni*^2^^+^
*> Cu*^2^^+^
*> Co*^2^^+^
*> Zn*^2^^+^
*> Cd*^2^^+^. 1-alkylimidazoles, as selective metal ion carriers, are used in the transportation of Cu(II), Zn(II), Co(II), and Ni(II) ions from nitrate and chloride acidic solutions through PIMs [72]. In both cases, the following order of separation has been discovered: *Cu*^2+^ > *Zn*^2+^ > *Co*^2+^ > *Ni*^2+^.

1,2,4-triazoles and their derivatives have also been studied as extractants of transition group metals. In case of Cu(II), the stability constant of 1-alkyl-1,2,4-triazoles depend on the length of the alkyl ligand chain and are in the following order: *C*_8_
*< C*_9_* < C*_10_
*< C*_11_* < C*_12_
*< C*_14_
*< C*_16_ [60]. In the case of extraction of nickel(II) ions, the order is as follows: *C*_8_
*> C*_9_
*> C*_10_
*> C*_14_ [73].

These examples of studies on the selective separation and extraction of ions of transition group metals from water solutions indicate the possibilities of applying various derivatives of azoles in extraction and membrane processes. Furthermore, these works also indicate the importance of the substituent applied to modify these compounds. In this study, a systemic analysis of the impact of the length of the alkyl substituent in 1,2,4-triazole on the ability of that compound to transport copper(II) ions through polymer inclusion membranes is carried out.

## 2. Materials and Methods

### 2.1. Reagents

In the study, *CuCl*_2_, CoCl_2_, NiCl_2_, dichloromethane CH_2_Cl_2_ (Chempur, Piekary Śląskie, Poland), NaCl (STANLAB, Lublin, Poland), cellulose triacetate (CTA, > 98%, Sigma-Aldrich, Poznań, Poland), and *o*-nitrophenyl octyl ether (ONPOE, Fluka, Buchs, Switzerland) were used. To synthesize 1-alkyl-1,2,4-triazoles (TRIA–n, n = 8, 9, 10, 11, 12, 14), the following compounds were used: sodium, 1,2,4-triazole and proper 1-bromoalkanes: 1-bromooctane, 1-bromononane, 1-bromodecane, 1-bromoundecane, 1-bromododecane (Sigma-Aldrich, Poznań, Poland), 1-bromotetradecane, and methanol (Chempur, Piekary Śląskie, Poland).

All reagents used were of analytical grade. Demineralised water was used to prepare water solutions (≤ 1 μS, HydroLab HLP Smart 1000 (HydroLab, Straszyn, Poland)).

### 2.2. Synthesis of 1-alkyl-1,2,4-triazole

1-alkyl-1,2,4-triazoles (n = 8, 9, 10, 11, 12, 14) used in the studies have been acquired from reaction of triazole with a proper 1-bromoalkane (Figure 1), in accordance to a procedure described above [74].

To prepare 1-alkyl-1,2,4-triazole, 75 mL of anhydrous methanol was added to a three-necked flask equipped with a stirrer, dropper, and backflow condenser, and then, 0.1 mol of metallic sodium was introduced gradually. When the sodium had fully reacted, 0.1 mol of 1,2,4-triazole was introduced into the flask. The contents were mixed thoroughly until the introduced triazole dissolved. Next, 0.1 mol of proper 1-bromoalkane was added into the flask. The reaction was carried out for 12 h at 65 °C and in the presence of the created sodium methanolate. Sodium bromide extracted in the course of the reaction was filtered out, and then, methanol was distilled out. The final product was distilled twice (1 Tor), yielding 1-alky-1,2,4-triazoles with approximately 70% efficiency. Distillation conditions of 1-alkyl-1,2,4-triazoles are presented in Table 1. The end product was crystalized from anhydrous ethyl acetate only in case of 1-tetradecyl-1,2,4-triazole.

Purity (99.9%) of acquired triazole derivatives were gauged with a gas chromatograph Hewlett Packard 5890 Series II equipped with a capillary column HP-5, flame ionization detector, and Hewlett Packard 3390A Integrator. The structures of 1-alkyl-1,2,4-triazoles were confirmed from ^1^H NMR and ^13^C NMR spectroscopy (Bruker ADVANCE, 400 Spectrometer, Germany) in DMSO-d6.


**1-octyl-1,2,4-triazole**


^1^H NMR (400 MHz, DMSO-*d*_6_, temp. 298K, TMS): δ [ppm] = 0.83–0.87(t, *J =* 7.0 MHz, 3H); 1.19–1.29(m, 10H); 1.75–1.79(t, J =7.2 MHz, 2H); 4.15-4.19(t, *J =* 7.0 MHz, 2H); 7.94(s, 1H); 8.51 (s, 1H).

^13^C NMR (100 MHz, DMSO-*d*_6_, temp. 298 K, TMS): δ [ppm] = 13.77; 21.96; 25.73; 28.29; 28.46; 29.18; 31.08; 48.45; 143.72; 151.13.


**1-decylo-1,2,4-triazole**


^1^H NMR (400 MHz, DMSO-*d*_6_, temp. 298 K, TMS): δ [ppm] = 0.84–0.87(t, *J =* 6.9 MHz, 3H); 1.17–1.27(m, 14H); 1.74–1.81(q, J = 5.4 MHz, 2H); 4.15–4.19(t, *J =* 7.1 MHz, 2H); 7.93(s, 1H); 8.51 (s, 1H).

^13^C NMR (100 MHz, DMSO-*d*_6_, temp. 298 K, TMS): δ [ppm] = 13.73; 22.01; 25.75; 28.37; 28.61; 28.83; 28.85; 29.20; 31.22; 48.45; 143.67; 151.09.


**1-undecylo-1,2,4-triazole**


^1^H NMR (400 MHz, DMSO-*d*_6_, temp. 298K, TMS): δ [ppm] = 0.84–0.87(t, *J =* 6.9 MHz, 3H); 1.19–1.29(m, 16H); 1.73–1.81(q, J = 7.2 MHz, 2H); 4.15–4.19(t, *J =* 7.1 MHz, 2H); 7.93(s, 1H); 8.51 (s, 1H).

^13^C NMR (100 MHz, DMSO-*d*_6_, temp. 298 K, TMS): δ [ppm] = 13.75; 22.02; 25.74; 28.37; 28.65; 28.83; 28.89; 28.92; 29.20; 31.24; 48.44; 143.69; 151.09.


**1-dodecylo-1,2,4-triazole**


^1^H NMR (400 MHz, DMSO-*d*_6_, temp. 298K, TMS): δ [ppm] = 0.84–0.87(t, *J =* 6.6 MHz, 3H); 1.24(m, 18H); 1.76–1.80(q, J = 7.2 MHz, 2H); 4.16–4.19(t, *J =* 7.1 MHz, 2H); 7.93(s, 1H); 8.51 (s, 1H).

^13^C NMR (100 MHz, DMSO-*d*_6_, temp. 298 K, TMS): δ [ppm] = 13.68; 22.03; 25.78; 28.42; 28.70; 28.87; 28.93; 29.00; 29.01; 29.23; 31.27; 48.45; 143.63; 151.05.


**1-tetradecylo-1,2,4-triazole**


^1^H NMR (400 MHz, DMSO-*d*_6_, temp. 298K, TMS): δ [ppm] = 0.83–0.87(t, *J =* 6.8, MHz, 3H); 1.19–1.27(m, 22H); 1.74–1.78(t, J = 7.2 MHz, 2H); 4.14–4.17(t, *J =* 7.0 MHz, 2H); 7.92(s, 1H); 8.49 (s, 1H).

^13^C NMR (100 MHz, DMSO-*d*_6_, temp. 298 K, TMS): δ [ppm] = 13.79; 22.02; 25.73; 28.35; 28.65; 28.81; 28.87; 28.95; 28.96; 28.98, 29.00; 29.18; 31.23; 48.44; 143.71; 151.12.

To measure the viscosity of the TRIA–n (n = 8, 9, 10, 11, 12, 14) a laboratory rheometer Physica MCR 501 (Anton Paar, Austria) was used. The shear rate was changed within the range of 1 to 700 s^−1^. A set of two cylinders were used in measurements (DG 26.7). To estimate the influence of temperature, the measurements were performed from 20 to 90 °C, with increments of 10 °C. All measurements were made using freshly distilled triazoles. Before the measurements, the samples were subjected to ultrasound homogenization.

### 2.3. Membrane Preparation

To synthesize PIMs, the following solutions were used: 12.5 g/dm^3^ CTA in dichloromethane, 10% *v*/*v* ONPOE in dichloromethane, and 0.10 mol/dm^3^ solution of the ion carrier (TRIA-n) in dichloromethane. Calculated volumes of the CTA solution, plasticizer, and carrier (in the ratio of CTA:ONPOE:TRIA-n = 4.5:1.5:1.5 *v*/*v*) were intensively mixed and this mixture was poured in to a Petri dish placed on a flat, horizontal surface. The surface area of the membrane thus obtained was 63.6 cm^2^. The dish was left to rest for 24 h for the solvent to evaporate in room temperature. Next, the membrane was weighed and its thickness was measured to an accuracy of ± 0.1 μm (PosiTector, DeFelsko, Ogdensburg, NY, USA). The thickness was measured at 17 different locations and was found to be 24.9 ± 0.9 μm. The carrier content was varied from 11.3 wt % to 15.8 wt % (Table 2). Each membrane was conditioned in demineralised water for 24 h before use.

### 2.4. Transport Experiment

A spiral membrane module was used in this study (Figure 2). These modules are characterized by a high quotient of membrane surface to volume of the feeding and receiving phases. This membrane module was also used by Schlosser to study the transport of carboxylic acids through supported liquid membranes (SLMs) [71,72]. Geometric parameters of the module are presented in Table 3. In every experiment, two module channel are clamped together. A membrane is located between two modules. Every two modules have an antagonistic spiral because clamping the spirals creates one channel separated by a membrane.

The membrane module was placed in a water bath. Tests (12 h) were performed at temperatures of 20 °C, 30 °C, 40 °C and 50 °C, accurate to within ± 0.5 °C. The feed and receiving phases were pumped from surge tanks using the module and then returned to the surge tanks. The flow rate of the feed and receiving phases was identical. The volume of both phases was the same (V = 25 cm^3^). The volumetric flow rate of both phases was kept constant at 3.0 cm^3^/min, except in the experiments for examining the influence of the volumetric flow rate.

The feed phase contained 0.1 mol/dm^3^ copper(II) chloride and a sufficient amount of chloride anions modified with NaCl. Overall, the feed phase contained equal amounts (0.1 mol/dm^3^) of *CuCl*_2_, CoCl_2_, and NiCl_2_. The concentration of the chloride anions was varied from 0.5 to 4.5 mol/dm^3^ in steps of 0.5 mol/dm^3^. The receiving phase was deionised water. Samples (0.2 cm^3^) for identifying the concentration of the transported metal ions in the receiving and feed phases were collected every 30 min for the first 4 h and every hour after that. The concentration in the feed phase was calculated using mass balance equations. To verify the accuracy of the analyses, the metal concentrations in the samples were measured every 2 h. Additionally, the concentration of the chloride anions was measured in the samples of the receiving phase. The pH of the feed phase was controlled to be approximately 5.5 throughout the experiment. Because the transport of ions through the studied PIM was not accompanied by the co-transport and counter-transport of protons, the pH of both phases did not need to be modified. The concentration of copper(II) in the samples containing only that cation was measured using a Shimadzu UV-VIS-2401 spectrometer (Shimadzu Europa, Duisburg, Germany) [75]. In case of mixed cations, the samples were analysed using an Agilent MP-AES 4200 emission spectrometer (Agilent, Santa Clara, CA, USA), utilising microwave-induced nitrogen plasma at a temperature of 5000 K to activate the elements. The concentration of the chloride anions in the receiving phase was measured using a Dionex DX-100 ion chromatograph equipped with a capillary column AS4SC, guard column AG4SC, and ASRS-II suppressor (DIONEX, Sunnyvale, CA, USA). Furthermore, 1.7 mM Na_2_CO_3_ + 1.8 mM NaHCO_3_ solution was used as the eluent.

Each of the samples were analyzed three times. For subsequent calculations, average values were considered. Experimental error, instrument error, and distribution of results considered as measurement uncertainty (standard deviation from the mean) did not exceed 3%. All measurement uncertainties were calculated using the Eurochem guide [76].

The initial rate of the transport of the metal ions through the PIM was assumed to be directly proportionate to the difference in metal ion concentrations in the feed and receiving phases. Based on this assumption, the concentration of the metal ions in the feed phase can be described using a first-order kinetics equation [77,78]
(1)$$\frac{dc}{dt}=-kc$$
where *c* (mol/dm^3^) is the concentration of the metal ions in the feed phase at a given time. In this equation, the rate constant *k* (1/s) considered all the processes occurring during ion transport.

The following dependency is obtained by integrating this equation with the initial condition *c* = *c*_0_ for *t* = 0
(2)$$\ln\left(\frac{c}{c_{0}}\right)=kt$$

This dependency is linear in the first stage of the experiment. Therefore, using linear regression analysis, the rate constant *k* can be determined.

Initial flux, *J*_0_ (mol/(m^2^s)), is the value characterising the transport of the metal ions through the PIM and can be expressed as
(3)$$J_{0}=\frac{V}{A}kc_{0}$$
where *V* (m^3^) is the volume of the feed phase and *A* (m^2^) is the area of the membrane.

Additionally, to quantitatively describe the process of ion separation during ion transport through the PIM, the selectivity coefficients, which are defined as the quotients of the initial metal ion flux *J*_0,*MI*_ and initial metal ion flux *J*_0,*MII*_, are calculated as
(4)$$S_{M_{I}/M_{II}}=\frac{J_{0,M_{I}}}{J_{0,M_{II}}},\;where\;J_{0,M_{I}}>J_{0,M_{II}}.\;$$
and the recovery coefficient, indicating the efficiency of metal ion separation from the feed phase, is defined as
(5)$$R=\frac{c_{0}-c}{c_{0}}\cdot 100\%$$


## 3. Results and Discussion

### 3.1. Influence of Volumetric Flow Rate of Phases on Transport of Cu(II) Ions

The effect of the volumetric flow rates of the feed and receiving phase passing through the membrane module on the hydrodynamic properties of the module is shown in Table 4.

Increase in the volumetric flow rate decreased the contact time of the phases in the module. As a result, the amount of copper(II) ions transported through the membrane decreased. However, the feed and receiving phases had a limited volume, and they operated in a closed circuit. Thus, each phase was pushed through the membrane module multiple times. This indicated that the increase in the volumetric flow rate increased the frequency with which these phases passed through the membrane module. This increase in circulation improved the work efficiency of the membrane module and thus increased the amount of copper(II) ions transported through the membrane. We observed two opposing processes; therefore, it was assumed that there was an optimal value for the volumetric flow rate that allowed for the largest quantity of copper(II) ions to be transported through the membrane.

The linear speed of the feeding and receiving phases is relatively low. As a result, the Reynolds number varies from about 80 to about 800. It is assumed that, for semicircular channels, the critical value of the Reynolds number for water is 1700 [79]. Below this value, as in the presented studies, the flow is laminar. In the presented results, no deposition of compounds on the membrane surface was observed. It can therefore be assumed that there is no effect of concentration polarization, typical for reverse osmosis [80].

In the experiment, the volumetric flow rate was varied from 0.5 to 4.5 cm^3^/min in steps of 0.5 cm^3^/min. The total concentration of the chloride anions was estimated on the level of 2 mol/dm^3^ with NaCl. TRIA-10 was used as the carrier.

When the volumetric flow rate of the feed and receiving phases was increased from 0.5 cm^3^/min to 2.0 cm^3^/min, the amount of transported copper(II) ions increased until it reached its maximum point (Figure 3). Further increase in the volumetric flow rate of the phases resulted in a decrease in the amount of copper(II) ions transported through the membrane. For volumetric flow rates above 3 cm^3^/min, the kinetic curves practically overlap.

The initial flux of the copper(II) ions changed in a similar manner (Figure 4). It increased, reached a maximum value at a volumetric flow rate of 2 cm^3^/min, entered a stage of oscillation, and finally stabilized at around 6.5 μmol/(m^2^s). This level was achieved with a flow rate of 3 cm^3^/min. When analyzing the obtained results, it should be concluded that the maximum value of Cu(II) transport for a flow rate of 2 cm^3^/min is the outlier result. If this result is skipped, it is clear that the increase in flow rate causes an increase in the initial flow of Cu(II) through the membrane until a plateau (Figure 4, dashed line). Its value is set at about 6.5 μmol/(m^2^s) for a flow of 3 cm^3^/min. Considering the above results, the flow rate of 3 cm^3^/min was considered for subsequent studies.

### 3.2. Chloride Anion Transport

Previous research on the transport of heavy metal ions through PIMs using 1-alkylimidazoles as carriers showed that this transport is selective toward NaCl [53,54]. As imidazoles and triazoles belong to the same group of chemical compounds, we can hypothesize that the transport of NaCl will not occur in case of alkyl derivatives of triazoles. To verify this hypothesis, research using TRIA-10 as the carrier has been conducted. The feed phase was a solution containing 0.1 mol/dm^3^ copper(II) and 2.0 mol/dm^3^ chloride. The concentration of chlorides in the samples of the receiving phase was analyzed. The results are presented in Figure 5.

All points representing the concentration of chloride anions lie on the straight line *y = a + bx*. Regression coefficients calculated together with the standard measurement uncertainty are *a =* (−1.24 ± 1.35) × 10^−4^ and *b =* 1.0215 ± 0.0100. As the confidence interval is 95% for the constant *a* equals (−4.1 × 10^−4^, 1.62 × 10^−4^) and contained the number 0, the intercept is statistically negligible. In such a situation, it is possible to establish the equation of the line going through the start of the coordinate system, *y =* (1.0142 ± 0.0061)*x*. The slope of this straight line is statistically significant and its value indicates that the quotient of [Cl^−^]/[Cu^2+^] is equal to 2. The mean value of this quotient calculated for all research points is 2.010 ± 0.015. Verification of the statistical null hypothesis that the expected value of this quotient is exactly 2.000 against the hypothesis that the value is higher than 2.000 allows for stating that at the significance level α = 0.05, there is no basis for rejecting it. Thus, it can be assumed that in every case, the concentration of chloride anions in the receiving phase is exactly two times greater than the concentration of the metal cations. This confirms our hypothesis that NaCl is not transported through the PIM when TRIA-10 is used as the carrier.

While investigating the transport mechanism through the PIM and the complex that the carrier forms with the transported chemical compounds, observations of the changes in the chlorides in the receiving phase may lead to varying conclusions. If the transport of ions through the PIM was simple, that is, it was not assisted by a chemical reaction, then both copper(II) and sodium(I) ions passed through the membrane, along with an equivalent number of chloride ions. A blind test using a membrane consisting only of CTA and ONPOE without a carrier showed that such membranes were an impenetrable barrier for copper(II) and sodium(I) ions. Considering this and the fact that the chloride concentration in the receiving phase was proportional to the copper(II) cation concentration, it could be assumed that the transport of ions through the PIM was assisted by a chemical reaction that is selective toward ions.

Lewis bases, such as azoles, consist of an atom with a lone electron pair and can create two types of complexes [81]: a neutral complex in the form of an adduct *ML_z_B_b_*, where Lewis base *B* creates a complex with a neutral, coordinatively unsaturated complex *ML_z_* (L–ligand, M–metal), and a complex in the form of an ion pair *ML_n_^(z−n)^(BH*^+^*)_n−z_*, where the anion complex *ML_n_^(z−n)^*, *n > z*, creates an extracted complex with the protonated Lewis base *BH^+^*.

Considering the NaCl + *CuCl*_2_ setup, these complexes could be created as a result of one of the following reactions:(6)$$Cu^{2+}+2Cl^{-}+nTRIA=CuCl_{2}TRIA_{n},\;\;\;n=1,2$$
(7)$$Cu^{2+}+nCl^{-}+\left(n-2\right)HTRIA^{+}=\left(CuCl_{n}^{2-n}\right)\cdot {\left(HTRIA^{+}\right)}_{n-2},\;\;\;n=2,3.\;$$

The experiment indicated that only *CuCl*_2_ passed -through the PIM, suggesting that the basic reaction accompanying the transport of copper(II) through the PIM was reaction (6). If we assumed that the transport was accompanied with reaction (7), that would mean that either *HCl* was transported through the membrane, or that we could observe the counter-transport of *HCl* in the form of a *HTRIA*^+^*Cl^−^* ion pair. In the first case, the previously obtained result (concentration of chlorides on the receiving side is twice the concentration of copper(II) ions) and the fact that there were no changes in the pH of either phase did not confirm this possibility. Simultaneously, the second option was ruled out because of the physical and chemical properties of alkyl triazole derivatives.

1-Alkyl-1,2,4,-triazoles are weak bases, significantly weaker than alkyl imidazole derivatives [82]. The pKa of an unsubituted 1,2,4-triazole is 2.50, while for its alkyl derivatives, this value ranges from 2.30 to 2.40 depending on the number of carbon atoms in the alkyl substituent, and the influence of the length of the substituent is small [59,68]. In case of 1,2,4-triazoles substituted in position 1, the donor of the coordination bond is a nitrogen atom in position 4 [83]. This nitrogen atom also undergoes protonation in an acidic environment [59,68]. However, considering that the transport takes place at a pH of about 5.5, protonation of the carrier can be ruled out. Thus, it can be concluded that a complex in form of an adduct *CuCl*_2_*TRIA_n_* is transported.

This conclusion agrees well with published results of studies on the structures of 1-alkyl-1,2,4-triazoles with transition group metals. Transition metal chlorides such as cobalt, nickel, and copper form complexes having a general form of *MCl*_2_*TRIA* or *MCl*_2_*TRIA*_2_ with 1-alkyl-1,2,4-triazoles, where *M = Cu, Co, Ni…* and *TRIA* = 1-alkyl-1,2,4-tirazole [84].

### 3.3. Influence of Concentration of Chloride Anions on Transport of Cu(II) Through PIM

Our previous research on the use of imidazole derivatives as carriers of metal ions through PIMs demonstrated an important effect of the concentration of chlorides in the feed phase on the efficiency and effectiveness of ion transport [53,54]. Keeping in mind that imidazoles and triazoles belong to the same group of chemical compounds—i.e., azoles—and that the manner in which these compounds react with transition group metals is similar, the concentration of chloride anions is expected to significantly impact the transport of copper ions through the PIM for alkyl triazole derivatives. Therefore, the influence of the concentration of chloride anions in the feed phase on the transport of copper(II) ions was tested. In this experiment, the concentration of chloride anions was varied from 0.5 to 5.0 mol/dm^3^ in steps of 0.5 mol/dm^3^. The concentration of copper(II) was 0.1 mol/dm^3^. The concentration of chloride anions was modified using NaCl. The pH of the feed phase was 5.5 and was kept constant throughout the experiment. The volumetric flow rate of the feed and receiving phases was 3.0 mol/dm^3^. TRIA-10 was used as the carrier.

Increasing the concentration of chloride anions in the feed phase significantly affected the amount of copper(II) ions being transported through the membrane (Figure 6, Table 5). Initially, the amount of copper(II) ions transported through the PIM increased with increasing concentration of chloride anions. The observed recovery coefficients after 6 and 12 h increased from 7.4% and 17.3% for a chloride concentration of 0.5 mol/dm^3^ to 15.1% and 28.7% for a chloride concentration of 1.5 mol/dm^3^, respectively. These were the maximum values of this parameter. Further increase in the chloride concentration caused a decrease in the recovery coefficients. At first, the recovery coefficient values were rather low and for a chloride concentration of 2 mol/dm^3^, the recovery coefficients after 6 and 12 h were 13.0% and 26.7%, respectively. However, further increase in the chloride concentration up to 5.0 mol/dm^3^ led to a significant decrease in the copper(II) ion recovery. At 5.0 mol/dm^3^, the recovery coefficients were 6.4% and 14.6% after 6 and 12 h, respectively, which were lower than the values obtained at a chloride concentration of 0.5 mol/dm^3^.

The initial speeds of transport through the PIM changed in a similar manner as the copper(II) recovery coefficients (Figure 7). The initial transport speed for a chloride concentration of 0.5 mol/dm^3^ was 3.97 μmol/(m^2^s). When the chloride concentration was increased to 1.5 mol/dm^3^, the initial speed of the copper(II) ions increased to 7.57 μmol/(m^2^s). Further increase in chloride concentration led to a decrease in the initial transport speed to 3.57 μmol/(m^2^s). Such speed corresponds to the chloride concentration of 0.5 mol/dm^3^.

In water solutions containing chloride anions, transition group metals like copper, nickel, cobalt, cadmium, and zinc can form stable chloride complexes [85,86,87,88,89,90,91]. In case of copper(II) ions, such complexes may be created in accordance with the reaction
(8)$$Cu^{2+}+iCl^{-}=CuCl_{i}^{2-i},\;\;\;\beta_{i}=\frac{\left[CuCl_{i}^{2-i}\right]}{\left[Cu^{2+}\right]{\left[Cl^{-}\right]}^{i}},\;\;\;i=1,2,3,4.\;$$

In chloride solutions, the concentration of *Cu*^2+^ decreases monotonically as the concentration of chloride anions increases, while the concentration of negative chloride complexes monotonically increases (Figure 8). In case of the positive chloride complex *CuCl*^+^, its concentration initially increases and starts decreasing after reaching its peak value. The neutral complex *CuCl*_2_ behaves in a similar manner. Contribution of this complex in the equilibrium mixture increases as the chloride concentration is increased to 1.5 mol/dm^3^, at which value it is at its peak. Its amount depends on the acidity of the solution and ionic strength, but the contribution of this complex in the solution is a few times greater than that of the remaining complexes.

**gure 8.** Distribution of copper(II) chloride complexes. Complexation constants of copper(II): log K_1_ = 0.06, log β_2_ = 0.67, log β_3_ = 0.20, and log β_4_ = −0.77 [85].

The point of maximal concentration of the neutral complex *CuCl*_2_ coincides with the maximum observed in results of the influence of the concentration of chloride anions on the transport of copper(II) ions through the PIM (Figure 5 and Figure 6). For this reason, it can be assumed that there is a connection between the distribution of copper(II) chloride complexes and the transport of ions through the PIM. This dependence additionally confirms that for the PIMs with alkyl 1,2,4-triazole derivatives serving as carriers, *Cu*_2_*TRIA_n_* complex created through reaction (6) is transported. The observed maximum transport and concentration of *CuCl*_2_ suggest assumption this concentration of chlorides for further research. However, because a chloride concentration lower than 1.5 mol/dm^3^ causes a rapid decrease in the concentration of the neutral complex *CuCl*_2_ and this decrease is much smaller for higher chloride concentrations, the chloride concentration of 2 mol/dm^3^ is considered for further research.

### 3.4. Influence of The Length of The Alkyl Ligand

In this study, alkyl chains of 1-alkyl-1,2,4-triazoles possessing 8, 9, 10, 11, 12 and 14 atoms of carbon were used. The volumetric flow rate of the feed and receiving phase was 3.0 cm^3^/min, while the concentration of chlorides in the feeding phase was 2.0 mol/dm^3^. The trials were performed at 20 °C.

Increase in the length of the carbon chain in the carrier causes a decrease in the amount of copper(II) ions transported through the PIM and the speed of the process (Figure 9). The highest values of the copper(II) ion recovery coefficient were achieved with TRIA-8 (Table 6). In this case, after 6 h, 32.0% of copper(II) ions permeate the membrane, while after 12 h, 50.3%. An increase in the length of the carbon chain by one carbon atom results in a decrease of the values of these parameters by about 50%. In such circumstances, the recovery coefficient for TRIA-9 is 15.8% and 27.6%, while for TRIA-10, the recovery coefficient is 14.0% and 26.4%, respectively, after 6 h and 12 h. Further increase in the length of the alkyl chain to 11 and 12 carbon atoms results in ~50% decrease of the recovery coefficients in comparison to those observed for TRIA-9 and TRIA-10. The lowest values of the recovery coefficient can be observed for TRIA-14, at 1.7% and 7.7%, respectively, after 6 h and 12 h. Analysing the results provided in Table 6, it is interesting that there is a possibility of combining carriers with 9 and 10, as well as 11 and 12 carbon atoms in pairs. Differences in recovery coefficients in these pairs are minimal, at only 1–2 percentage points.

A similar situation as for the recovery coefficients can be observed for the values of initial copper(II) ion streams going through the PIM (Figure 10). The highest initial flux (16.1 μmol/(m^2^s)) is observed in case of TRIA-8. Increase in the length of the carbon chain in the alkyl substituent by 1–2 carbon atoms results in over 50% of reduction in the initial ion transport speed through the PIM. The observed initial values are equal to 7.44 and 6.80 μmol/(m^2^s), respectively, for TRIA-9 and TRIA-10 carriers. Further increase in the length of the ligand by 1 or 2 carbon atoms results in a further decrease of these values to 4.00 and 3.16 μmol/(m^2^s) respectively for TRIA-11 and TRIA-12, again by about 50%. In case of 1-tetradecyl-1,2,4-triazole, the initial speed of copper(II) transport through the PIM is 1.59 μmol/(m^2^s), which is about 50% lower than its counterpart for TRIA-12.

Differences observed in copper(II) transport through the PIMs can be associated with differences in the viscosity of individual 1-alkyl-1,2,4-triazoles homologues (Table 1). TRI-8 has the lowest viscosity (6.834 mPa·s). The viscosities of TRIA-9 and TRIA-10 are higher by about 25% in comparison to TRIA-8, and equal to 8.464 and 8.609 mPa·s respectively. The viscosity of another homologue—TRIA-11, equals 12.57 mPa·s, and is about 50% higher than the viscosities of homologues with alkyl chains containing 9 and 10 carbon atoms. TRIA-14 is solid with a melting point of 47–49 °C; hence; it is assumed that this was its form in the tests in this study. TRIA-12 is greasy, owing to which the viscosity measurements were not successful.

With such changes to viscosity or the states of 1-alkyl-1,2,4-triazoles applied in this work as carriers of copper(II) ions, the observed changes in the initial speeds correlate. TRIA-9 and TRIA-10 have comparable viscosities and comparable initial transport speeds, which are lower than for TRIA-8. On the other hand, TRIA-11 has a significantly higher viscosity compared to other carriers; however, the initial speed of transport is significantly lower. An additional component to consider is the viscosity of ONPOE, the applied plasticiser, that equals 11.8 mPa·s. The viscosities of TRIA-8, 9, and 10 are significantly lower than that of ONPOE, while the viscosity of TRIA-11 is higher (Table 1).

Results of research in which TRIA-12 and TRIA-14 have been applied as carriers of copper(II) ions require separate clarifications. In these two cases, ONPOE, when used as the plasticiser, serves as a solvent for applied carriers [92]. Therefore, transport of copper(II) ions through the PIM occurs, despite the fact that the state of these carriers does not suggest it. However, initial fluxes are very low.

Differences in the transport speed described here may encourage the question about the mechanism of transport through the PIMs. In literature concerning this subject, two basic transport mechanisms are distinguished: diffusion and jump [93,94]. In the former, the complex of the carrier and transported chemical compound diffuses through the membrane, using the plasticiser as the liquid phase. In the latter, the carrier is embedded into the structure of the membrane, while the transported chemical compound jumps from one molecule of the carrier to another. Data presented indicate the diffusion mechanism. However, this requires additional research that exceeds the scope of this work.

### 3.5. Temperature Effect

The activation energy is a significant parameter characterising the transport of ions through the PIM membranes. It can be determined using the Eyring equation, which describes the dependencies between the transport speed through the PIM and temperature [95,96,97].
(9)$$\ln J_{0}=\ln J_{\infty }-\frac{E_{a}}{R}\cdot \frac{1}{T}$$
where: *J*_0_*, mol/(m*^2^*s)* is the initial flux trough PIM, *J_∞_* is the initial flux at infinitely high temperature, *E_a_, J/mol* is the activation energy, *R* = 8.314 *J*/(*mol K*) is the ideal gas constant, and *T*, *K* is the temperature.

To determine the activation energy, tests at temperatures of 20, 30, 40, and 50 °C were performed, with the following parameters: Flow rate, 3.0 cm^3^/min; concentration of chlorides, 2.0 cm^3^/min; carrier, TRIA-10.

Increase in the temperature during transport causes a significant increase in the amount of copper(II) ions transported through the PIM and the speed at which this occurs (Figure 11). Recovery coefficients (Table 7) after 6 and 12 h of the process equal 12.9% and 26.4% at 20 °C, while for 50 °C, they are 87.6% and 96.5%, respectively.

Increase in the angle of inclination of individual kinetic curves with respect to the OX axis with increasing temperature indicates an increase in the initial speed of transport through the PIM (Figure 12). The initial flux at 20 °C is low, at 6.29 μmol/(m^2^s). Increasing the temperature by 10 °C results in this parameter increasing almost 2.5 times, up to 14.4 μmol/(m^2^s). A further increase by 10 °C increases the initial speed by another 2.5 times, up to 36.6 μmol/(m^2^s). During transport through the PIM at 50 °C, the initial speed is 94.5 μmol/(m^2^s), which is over 2.5 times higher than for transport at 40 °C and 15 times higher than for transport at 20 °C.

The increase in the initial transport speed through the PIMs can be explained by some observable phenomena. Firstly, increase in the temperature causes a decrease in the viscosity of the plasticizer and carrier (Figure 13), and an increase in the mobility of CTA chains in the polymer matrix. Such an increase in the mobility of the polymer chains associated with the decrease in the viscosity of other elements of the membrane will facilitate the migration of the plasticizer and the carrier dissolved in it, deeper within the membrane. Hence, the speed at which copper(II) is transported through the PIM will increase. However, the largest influence on the temperature-related speed of transport by PIM is the increase in the speed of the chemical reaction in which the complex transported through the membrane is created. Hence, temperature has a significant influence on the speed of transport, as diffusion processes are significantly less susceptible to the impact of temperature than processes related to chemical reaction.

Trials performed at different temperatures enabled the use of the Eyring equation to calculate the activation energy (Figure 14). Points corresponding to individual values of the initial copper(II) flux through the PIM lie on a straight line. The equation obtained is statistically significant, while the correlation coefficient R^2^ equals 0.9964 and indicates the significance of the relation between temperature changes and *ln*(*J*_0_). With the slope of the straight line, the activation energy was calculated to be 8575.5 ± 366.1, which is also statistically significant. Thus, the activation energy *E_a_* = (71.3 ± 3.0) kJ/mol. Such a high value of activation energy suggests that the transport of copper(II) ions through the PIMs containing 1-alkyl-1,2,4,-triazoles is controlled kinetically by the reaction of copper(II) with the carrier.

The determined activation energy is higher than that obtained in previous works. Salazar-Alvarez [95], while studying lead(II) transport through PIMs with di-2-ethylhexyl phosphoric acid as the carrier, tris-(2-butoxyethyl)-phosphate as the plasticizer, and CTA as the polymer matrix, determined the activation energy to be 11 kJ/mol. Fontas [97], who studied the transport of platinum(IV) and cadmium(II), determined the activation energies to be 17.3 and 17.8 kJ/mol, respectively. In his study, he used PIMs with CTA as the matrix, Aliquat 336 and Lasalocid A as the ion carriers, and ONPOE as the plasticizer. Kozłowski [98], while the studying transport of chromium(VI) through the PIM (TOA, ONPPE, CTA), determined the activation energy to be 30.5 kJ/mol. A similar activation energy (30.65 kJ/mol) was calculated by Chaudry [99], who studied chromium(VI) ion transport through SLM with TOA as the carrier. In each of these cases, the authors interpreted the determined activation energies as confirmations of control of transport by diffusion.

Comparison of the activation energy obtained in this work with those obtained by other researchers poses certain methodological problems. Firstly, environments in which the tests were carried out are incomparable. Secondly, it is difficult to compare extractants that differ vastly in nature, in terms of their chemical properties (acidic, neutral, and basic extractants), and of the complexes that they form (chelate complexes, ion pairs, or adducts) with the ions they transport. The results of Kozłowski [98] and Chaundry [99], who used TOA as carriers in membranes, appear most similar to those presented in this work. Despite their high values of activation energy (over 30 kJ/mol), the authors interpreted it as confirmation of the control of transport by diffusion. It seems more appropriate to assume that these results indicate on an intermediate area, controlled by both the diffusion of the complex through the membrane and the reaction in which the complex is created. Such reasoning is consistent with suggestions of the authors, who assumed that activation energies exceeding 50 kJ/mol indicate transport controlled by a chemical reaction, while those below 30 kJ/mol indicate transport controlled by diffusion [100]. Thus, one can conclude that the activation energy values quoted here show how the transport mechanism shifts from one controlled by diffusion to one controlled by a chemical reaction, following changes in the ion carrier from typically acidic (D2EHPA), through basic (TOA), and finally to neutral (TRIA-n). The complex structure is also considered; it changes from chelate, through ion pair, to adduct.

### 3.6. Separation

In many waste solutions, copper coexists with nickel and cobalt. Therefore, tests to achieve the equimolar separation of mixtures containing not only copper(II) ions, but also ions of nickel(II) and cobalt(II), were carried out. The feeding phase was a solution containing 0.1 mol/dm^3^ ions of each metal. The total concentration of chloride anions was 2 mol/dm^3^. In the trials, membranes containing TRIA-8, TRIA-10, and TRIA-12 were used as ion carriers. Exact contents of the membranes are presented in Table 8.

The volumetric flow rate is 3.0 cm^3^/min. Feeding phase: MCl_2_ concentration—0.1 mol/dm^3^, chloride concentration—2.0 mol/dm^3^. Receiving phase: demineralized water. Membrane: CTA–ONPOE–TRIA-10. The composition of the membrane is provided in Table 2. Temperature: 20 ± 0.5 °C.

Initial fluxes of metal cations (Table 8) decrease as the alkyl substituent (*TRIA-8 > TRIA-10 > TRIA-12*) length increases. Analysis of the initial fluxes of copper(II), nickel(II) and cobalt(II) ions through the PIMS with TRIA-n as carriers indicates a sequence of separation, of *Cu(II) > Ni(II) > Co(II)*. The selectivity coefficients exhibit the same trend. High values of the selectivity coefficients of copper(II) to nickel(II) and cobalt(II) indicate a possibility of separating the copper(II) ions from such mixtures. Hence, separation of ions to cations should be possible. The selectivity coefficient of nickel(II) to cobalt(II) is also significantly high; hence, it should be possible to separate those cations even here. Notably, because nickel and cobalt often coexist and exhibit similar physical and chemical properties, it is difficult to separate them with extraction methods.

The results obtained in this study indicate a clear advantage in the selectivity of 1-alkyl-1,2,4-triazoles in relation to Cu^2+^ ions. A similar result was obtained in other studies where azole derivatives were used [70,71,72]. However, in the case of Ni^2+^ and Co^2+^ ions, the selectivity tests are not so obvious. Some authors give the order as Ni^2+^ > Co^2+^ [70,71], while others the opposite [72]. It seems that these differences are due on the one hand to the great chemical similarity of these ions, and on the other hand, to the presence of additional functional groups such as pyridyl or aldoxime.

## 4. Conclusions

The results of this study demonstrate the possibility of employing 1-alkyl-1,2,4-triazoles (TRIA-n, n = 8, 9, 10, 11, 12, 14) as selective carriers of copper(II), nickel(II), and cobalt(II) ions through PIMs containing ONPOE as the plasticiser and CTA as the polymer matrix. In the first stage of the research, the possibility of NaCl transport through the matrix was ruled out by proving that the concentration of chloride anions on the receiving side is twice greater than the concentration of copper(II) ions. Tests indicated that the transport speed through the PIM is largely dependent on the volumetric flow rate of the phases through the membrane module, and that it is possible to determine its optimal value. The optimal value at which the maximum transport speed of the ions through the PIM and rate of copper(II) ion recovery were observed, was 2–3 cm^3^/min. While studying the influence of the concentration of chloride anions in the feeding phase, the best results were obtained in the concentration range of 1.5–2.0 mol/dm^3^. The observed maximum coincides with the maximal concentration of *CuCl*_2_ in the feeding phase, which may be indirect evidence that a complex in the form of *CuCl*_2_TRIA_n_, n = 1,2 participates during ion transport through the studied PIMs. Application of other homologues of 1-alkyl-1,2,4-triazole demonstrates a significant impact of the alkyl ligand on the selective separation of copper(II) ions from the ternary solution containing additional ions of nickel(II) and cobalt(II). The obtained results indicate that TRIA-8 is the metal ion carrier with the best properties, and suggest that the viscosity of the plasticiser and the carrier has a significant influence on the process; this is demonstrated by tests performed at different temperatures. The transport speed increased as the temperature increased, owing to the decrease in viscosity and related increase in the mobility of the complex moving through the membrane, as well as due to the increase in metal ions complexing on the boundaries of the two phases, namely, the feeding phase/membrane and membrane/receiving phase. The significant importance of the speed of chemical reactions is evinced by the high activation energy, i.e., 71.3 ± 3.0 kJ/mol. Determination of the mechanism of metal ion transport through the PIMs containing 1-alkyl-1,2,4-triazoles and the stage limiting the speed of the transport requires further research

## Figures and Tables

**Figure 1 membranes-10-00201-f001:**
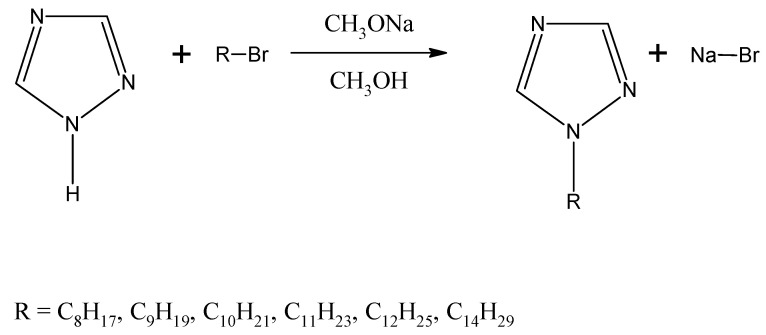
Scheme of alkylation reactions of triazoles with bromoalkanes.

**Figure 2 membranes-10-00201-f002:**
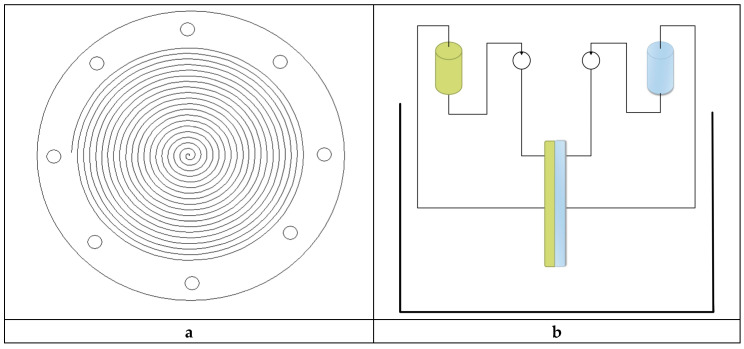
(**a**) PTFE module with Archimedean spiral. (**b**) Scheme of equipment.

**Figure 3 membranes-10-00201-f003:**
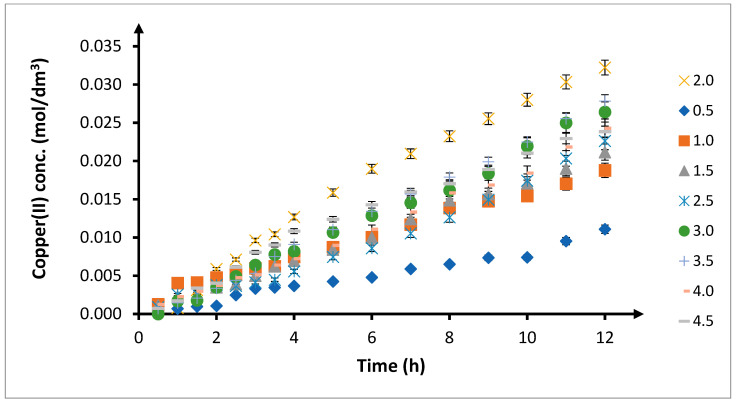
Change in concentration of copper(II) in receiving phase over time depending on volumetric flow rate of phase in spiral module. Flow rate: 0.5–4.5 cm^3^/min; feed phase: *CuCl*_2_ concentration = 0.1 mol/dm^3^, chloride concentration = 2.0 mol/dm^3^; receiving phase: demineralized water; and membrane: CTA-ONPOE-TRIA-10. The composition of the membrane is given in Table 2. Temperature: 20 ± 0.5 °C.

**Figure 4 membranes-10-00201-f004:**
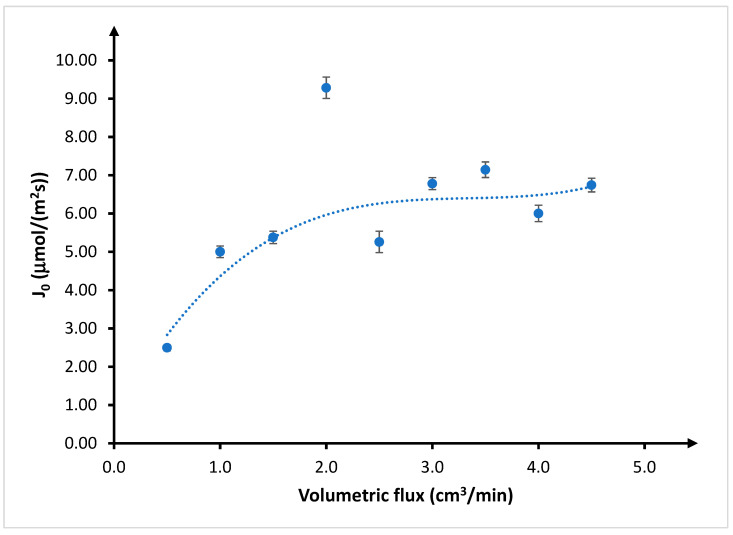
Initial copper(II) flux (J_0_, μmol∙m^−2^∙s^−1^) depending on volumetric flow rate of feed and receiving phases. Feed phase: *CuCl*_2_ concentration = 0.1 mol/dm^3^, chloride concentration = 2.0 mol/dm^3^; receiving phase: demineralized water; and membrane: CTA-ONPOE-TRIA-10. The composition of the membrane is given in Table 2. Temperature: 20 ± 0.5 °C.

**Figure 5 membranes-10-00201-f005:**
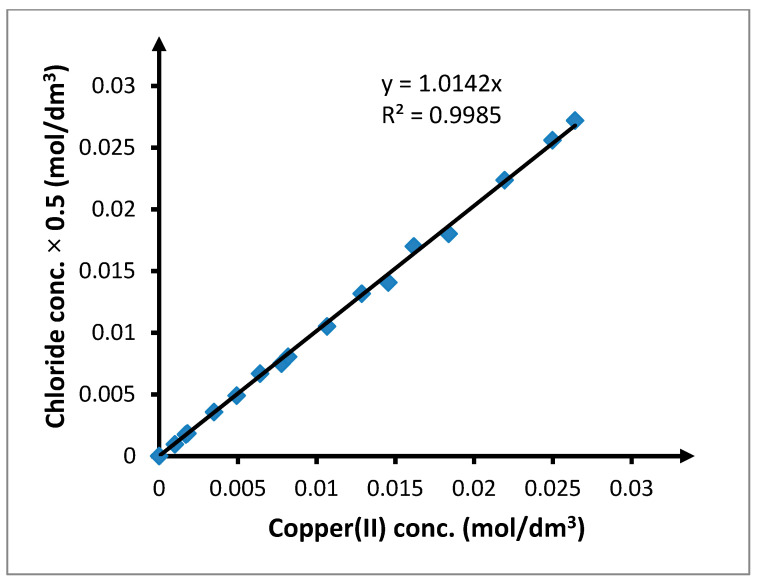
Relationship between chloride anion concentration and copper(II) concentration in receiving phase. The volumetric flow rate was 3.0 cm^3^/min. Feed phase: *CuCl*_2_ concentration = 0.1 mol/dm^3^, chloride concentration = 2.0 mol/dm^3^; receiving phase: demineralized water; and membrane: CTA-ONPOE-TRIA-10. The composition of the membrane is given in Table 2. Temperature: 20 ± 0.5 °C.

**Figure 6 membranes-10-00201-f006:**
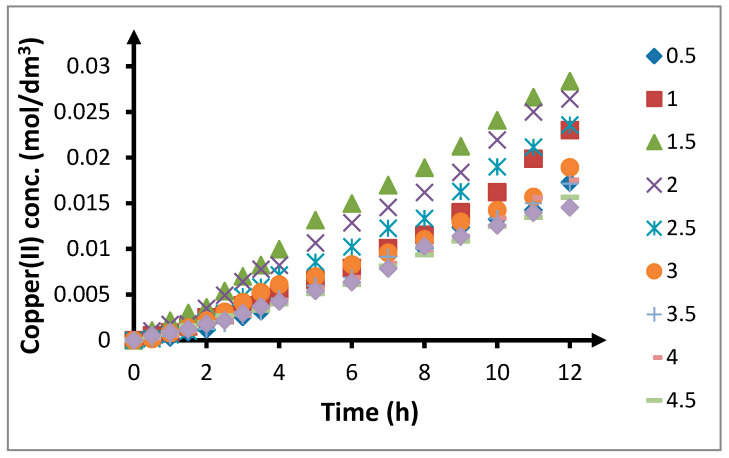
Change in concentration of copper(II) in receiving phase over time depending on concentration of chloride anions in feed phase. The volumetric flow rate is 3.0 cm^3^/min. Feed phase: *CuCl*_2_ concentration = 0.1 mol/dm^3^, chloride concentration = 0.5–5.0 mol/dm^3^; receiving phase: demineralized water; and membrane: CTA-ONPOE-TRIA-10. The composition of the membrane is given in Table 2. Temperature: 20 ± 0.5 °C.

**Figure 7 membranes-10-00201-f007:**
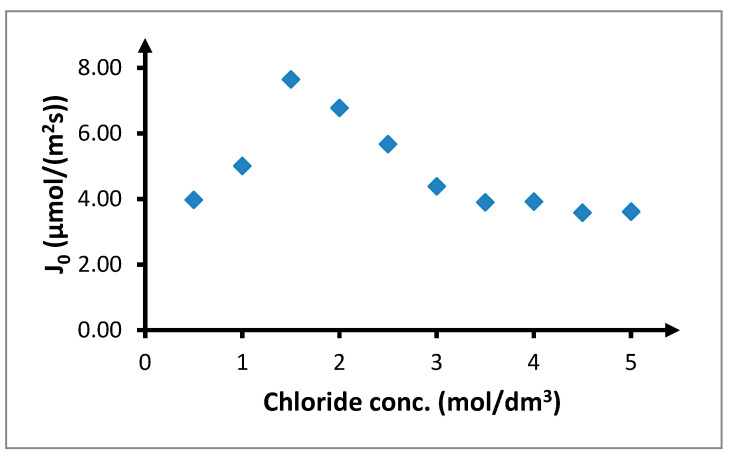
Initial copper(II) flux (*J*_0_, μmol∙m^−2^∙s^−1^) depending on concentration of chloride anions in feed phase. The volumetric flow rate is 3.0 cm^3^/min. Feed phase: *CuCl*_2_ concentration = 0.1 mol/dm^3^, chloride concentration 0.5 = 5.0 mol/dm^3^; receiving phase: demineralized water; and membrane: CTA-ONPOE-TRIA-10. The composition of the membrane is given in Table 2. Temperature 20 ± 0.5 °C.

**Figure 8 membranes-10-00201-f008:**
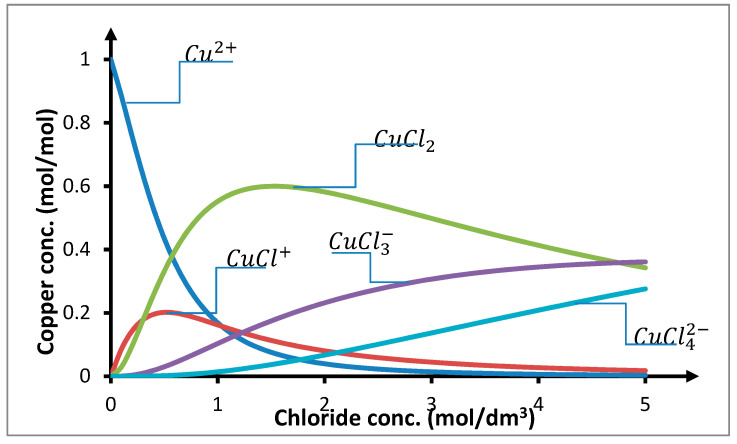
Distribution of copper(II) chloride complexes. Complexation constants of copper(II): log K_1_ = 0.06, log β_2_ = 0.67, log β_3_ = 0.20, and log β_4_ = −0.77 [85].

**Figure 9 membranes-10-00201-f009:**
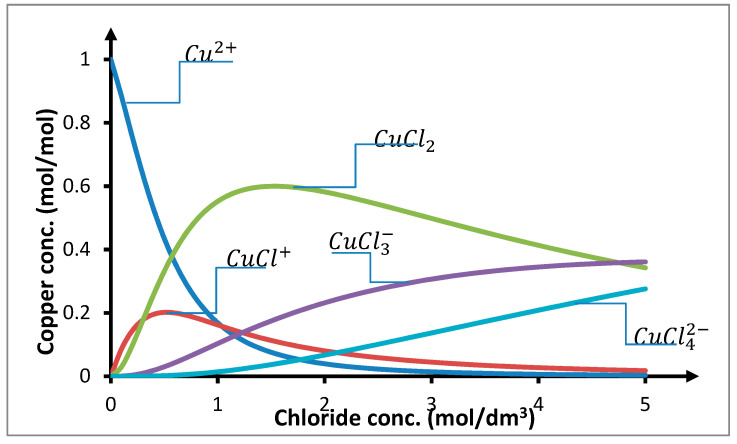
Effect of the chain length of the alkyl substituent on the transfer of copper(II) ions by PIM. The volumetric flow rate is 3.0 cm^3^/min. Feeding phase: *CuCl*_2_ concentration—0.1 mol/dm^3^, chloride concentration—2.0 mol/dm^3^. Receiving phase: demineralized water. Membrane: CTA-ONPOE-TRIA-10. The composition of the membrane is provided in Table 2. Temperature 20 ± 0.5 °C.

**Figure 10 membranes-10-00201-f010:**
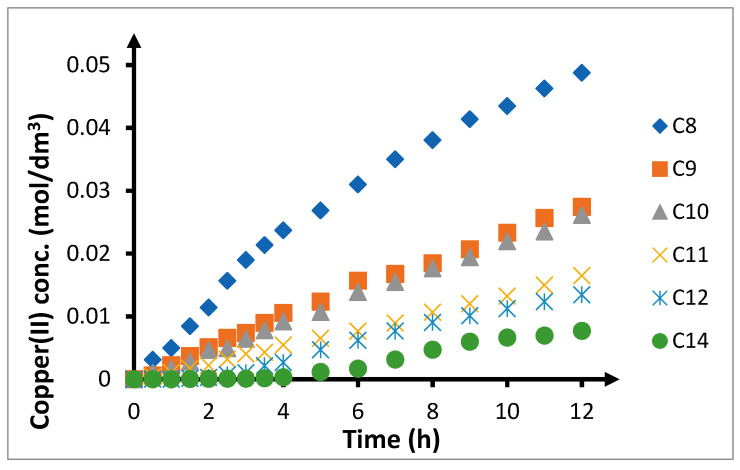
Effect of the chain length of the alkyl substituent on the initial flux of copper(II) ions. The volumetric flow rate is 3.0 cm^3^/min. Feeding phase: *CuCl*_2_ concentration—0.1 mol/dm^3^, chloride concentration—2.0 mol/dm^3^. Receiving phase: demineralized water. Membrane: CTA-ONPOE-TRIA-10. The composition of the membrane is provided in Table 2. Temperature 20 ± 0.5 °C.

**Figure 11 membranes-10-00201-f011:**
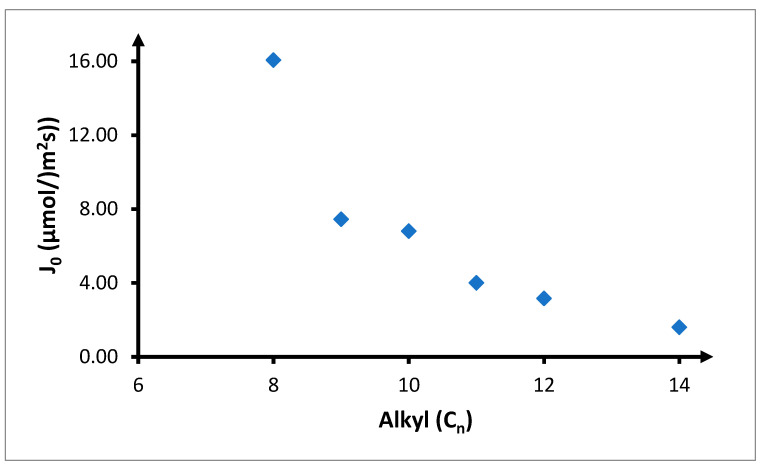
Influence of temperature on the transfer of copper(II) ions by PIM. The volumetric flow rate is 3.0 cm^3^/min. Feeding phase: *CuCl*_2_ concentration—0.1 mol/dm^3^, chloride concentration—2.0 mol/dm^3^. Receiving phase: demineralized water. Membrane: CTA-ONPOE-TRIA-10. The composition of the membrane is provided in Table 2.

**Figure 12 membranes-10-00201-f012:**
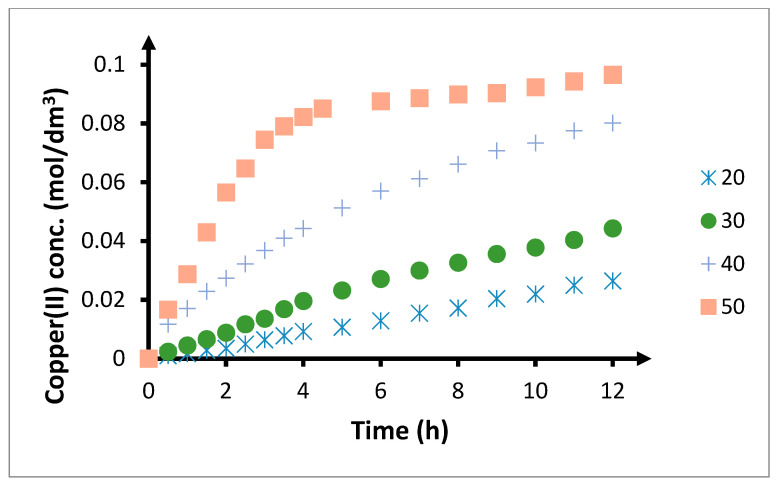
Effect of temperature on the initial flux of copper(II) ions. The volumetric flow rate is 3.0 cm^3^/min. Feeding phase: *CuCl*_2_ concentration—0.1 mol/dm^3^, chloride concentration—2.0 mol/dm^3^. Receiving phase: demineralized water. Membrane: CTA-ONPOE-TRIA-10. The composition of the membrane is provided in Table 2.

**Figure 13 membranes-10-00201-f013:**
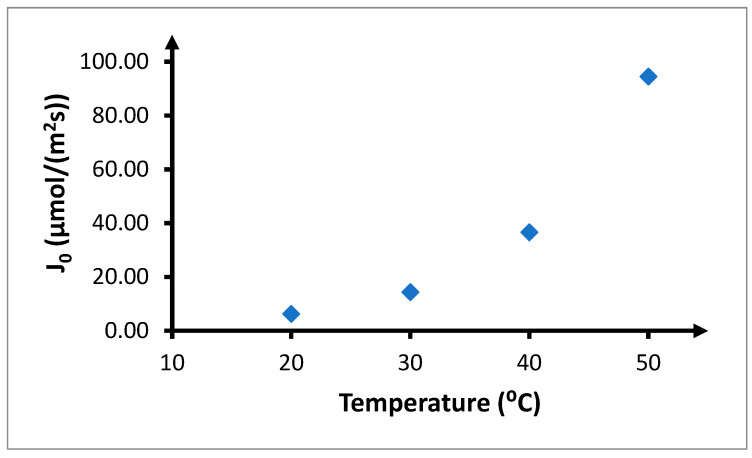
Viscosity dependence of TRIA-n (n = 8, 9, 10, 11) and ONPOE on temperature.

**Figure 14 membranes-10-00201-f014:**
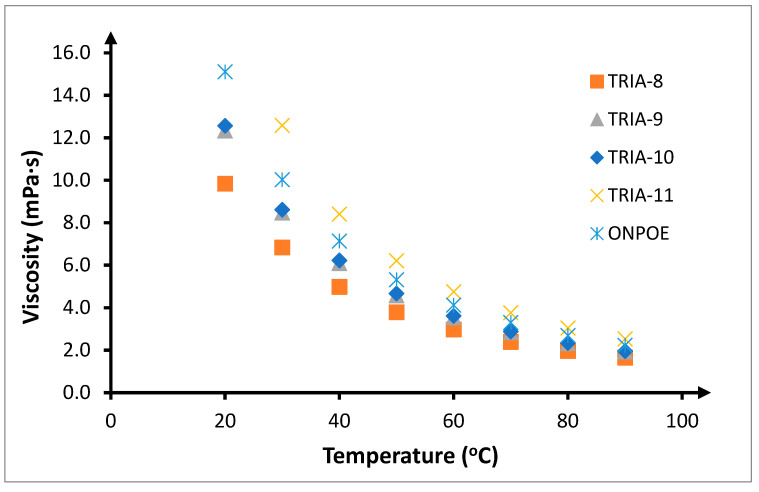
Graphical representation of the Eyring equation. The volumetric flow rate is 3.0 cm^3^/min. Feeding phase: *CuCl*_2_ concentration—0.1 mol/dm^3^, chloride concentration-2.0 mol/dm^3^. Receiving phase: demineralized water. Membrane: CTA-ONPOE-TRIA-10. The composition of the membrane is provided in Table 2.

**Table 1 membranes-10-00201-t001:** Properties of 1-alkyl-1,2,4-triazoles

Compound	Distillation Parameters		Yield (%)	Viscosity * (mPa s)
	Pressure (bar) (hPa)	Boiling Temp. (°C)		
1-oktylo-1,2,4-triazole	1	185–188	78	6.834
1-nonylo-1,2,4-triazole	1	201–203	70	8.464
1-decylo-1,2,4-triazole	1	216–218	75	8.609
1-undecylo-1,2,4-triazole	1	225–226	68	12.57
1-dodecylo-1,2,4-triazole	1	239–241	73	-
1-tetradecylo-1,2,4-triazole	melting point = 47–49 °C		66	-

* Measurement made at 30 °C.

**Table 2 membranes-10-00201-t002:** Membrane composition (wt %).

Carrier	Content (wt %)		
	TRIA-n	ONPOE	CTA
TRIA-8	11.3	65.2	23.5
TRIA-9	12.1	64.6	23.3
TRIA-10	12.9	64.0	23.1
TRIA-11	13.6	63.5	22.9
TRIA-12	14.3	63.0	22.7
TRIA-14	15.1	62.4	22.5

**Table 3 membranes-10-00201-t003:** Geometric properties of spiral module.

Module Diameter	120 mm
Membrane diameter	90 mm
Module thickness	10 mm
Outer diameter of spiral	77.0 mm
Channel depth	0.25 mm
Channel width	1.42 mm
Channel length	239.2 cm
Effective channel area	33.96 cm^2^
Channel volume	0.849 cm^3^

**Table 4 membranes-10-00201-t004:** Hydrodynamic properties of spiral module.

Volumetric Flux (cm^3^/min)	Rate oAqueous Phase Exchange in 1 h	Contact Time (s)	Linear Speed (m/s)	Reynolds Number
0.5	1.2	105.60	0.023	87
1.0	2.4	52.80	0.047	177
1.5	3.6	35.20	0.070	263
2.0	4.8	26.40	0.094	354
2.5	6.0	21.12	0.117	440
3.0	7.2	17.60	0.141	531
3.5	8.4	15.09	0.164	617
4.0	9.6	13.20	0.188	708
4.5	10.8	11.73	0.211	794

Feed phase: *CuCl*_2_ concentration = 0.1 mol/dm^3^, chloride concentration = 2.0 mol/dm^3^; receiving phase: demineralized water; and membrane: CTA-ONPOE-TRIA-10. The composition of the membrane is given in Table 2. Temperature: 20 ± 0.5 °C.

**Table 5 membranes-10-00201-t005:** Recovery coefficient (R%) of Cu(II) ions after 6 and 12 h depending on concentration of chloride anions in feed phase.

Cl^-^ (mol/dm^3^)	Recovery Coefficient, R (%)	
	6 h	12 h
0.5	7.4	17.3
1.0	8.0	23.1
1.5	15.1	28.7
2.0	13.0	26.7
2.5	10.2	23.6
3.0	8.3	19.0
3.5	6.9	17.2
4.0	6.9	17.6
4.5	6.1	15.7
5.0	6.4	14.6

The volumetric flow rate is 3.0 cm^3^/min. Feed phase: *CuCl*_2_ concentration = 0.1 mol/dm^3^, chloride concentration = 0.5–5.0 mol/dm^3^; receiving phase: demineralized water; and membrane: CTA-ONPOE-TRIA-10. The composition of the membrane is given in Table 2. Temperature: 20 ± 0.5 °C.

**Table 6 membranes-10-00201-t006:** Recovery coefficient (R, %) of Cu(II) ions after 6 h and 12 h (R, %) depending on the length of the alkyl substituent.

Carrier	Recovery Coefficient, R (%)	
	6 h	12 h
TRIA-8	32.0	50.3
TRIA-9	15.8	27.6
TRIA-10	14.0	26.4
TRIA-11	7.7	16.5
TRIA-12	6.2	13.4
TRIA-14	1.7	7.7

The volumetric flow rate is 3.0 cm^3^/min. Feeding phase: *CuCl*_2_ concentration—0.1 mol/dm^3^, chloride concentration—2.0 mol/dm^3^. Receiving phase: demineralized water. Membrane: CTA-ONPOE-TRIA-10. The composition of the membrane is provided in Table 2. Temperature: 20 ± 0.5 °C.

**Table 7 membranes-10-00201-t007:** Recovery coefficient (R, %) of Cu(II) ions after 6 h and 12 h depending on the temperature.

Temperature (°C)	Recovery Coefficient, R (%)	
	6 h	12 h
20	12.9	26.4
30	27.1	44.3
40	57.0	80.1
50	87.6	96.5

The volumetric flow rate is 3.0 cm^3^/min. Feeding phase: *CuCl*_2_ concentration—0.1 mol/dm^3^, chloride concentration—2.0 mol/dm^3^. Receiving phase: demineralized water. Membrane: CTA-ONPOE-TRIA-10. The composition of the membrane is provided in Table 2.

**Table 8 membranes-10-00201-t008:** Initial flux and selectivity coefficients for the competitive transport of Cu (II), Ni (II), and Co (II) ions by PIMs with TRIA-n (n = 8,10,12) as a carrier

Metal Ions	Carrier					
	TRIA-8		TRIA-10		TRIA-12	
	J_0_ (μmol/(m^2^s))	$S_{\frac{M_{I}}{M_{II}}}$	J_0_ (μmol/(m^2^s))	$S_{\frac{M_{I}}{M_{II}}}$	J_0_ (μmol/(m^2^s))	$S_{\frac{M_{I}}{M_{II}}}.\;$
Cu(II)	17.3 ± 1.2	$Cu\begin{array}{c} >\\ 46.1 \end{array}Ni\begin{array}{c} >\\ 12.4 \end{array}Co$.	9.9 ± 0.5	$Cu\begin{array}{c} >\\ 32.0 \end{array}Ni\begin{array}{c} >\\ 6.31 \end{array}Co$	1.90 ± 0.10	$Cu\begin{array}{c} >\\ 33.5 \end{array}Ni\begin{array}{c} >\\ 2.64 \end{array}Co$
Ni(II)	0.376 ± 0.003		0.308 ± 0.008		0.0052 ± 0.0005	
Co(II)	0.0304 ± 0.0017		0.049 ± 0.003		0.00215 ± 0.000	

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
