# Peer review of "Facilitated Transport of Copper(II) across Polymer Inclusion Membrane with Triazole Derivatives as Carrier"

_membranes, 2020, doi:10.3390/membranes10090201_

Round 1

Reviewer 1 Report

The article contains interesting material concerning the study of ion-selective properties of liquid membranes of the PIM type. At first stage, 1-alkyl-1,2,4-triazoles (n = 8, 9, 10, 11, 12, 14) using in the study have been obtained by reaction of triazole with a proper 1-bromoalkane. At second one, PIM membrane based cellulose triacetate was made. Triazoles are used by the authors as carriers of charged particles, due to which selective transport of metal ions through the membrane is carried out. The problem statement is related to the analysis of the available literature data on the differences in the rates of transfer of ions of the same type through membranes containing triazoles with alkyl fragments of different lengths. The authors attempt to study this effect using the example of separated mixtures containing bivalent copper ions. This part of the work is the main one and seems to be the most interesting. Triazoles are used by the authors as carriers of charged particles, due to which selective transport of metal ions through the membrane is carried out. The problem statement is related to the analysis of the available literature data on the differences in the rates of transfer of ions of the same type through membranes containing triazoles with alkyl fragments of different lengths. The authors attempt to study this effect using the example of separated mixtures containing bivalent copper ions. This part of the work is the main one and seems to be the most interesting.

At the same time, the article has many minor flaws. On the subject and content, the manuscript is fully suitable for publication in Membranes. I would like to find in the article the assumptions about the reason for the effect of the alkyl fragment in the carrier molecule on the transport properties of membranes. Many minor flaws are associated with the fact that the author was in a hurry to present the material, as well as with the problem of presentation in English (there are contextual inaccuracies noted in the attached file).  Designations and abbreviations, such as TRIA, MLn(z-n)(BH+)n-z, B and others, should be entered in a separate list and shown in the article where they are introduced for the first time. Table 8 is difficult to read and requires explanations. If the shortcomings of the presentation are improved, the work will win and will be interesting to the reader.

Reviewer 2 Report

The study focused on the transport of copper(II) ion across polymer inclusion membrane (PIM) with triazole comprising different carbon chain length. The results are organized well. The following are my concerns:

  1. In page 3, line 96-102, the authors may also mention why the other works have a different order of separating ions? What is the major factor affecting this trend?

  2. Why there is no viscosity result for 1-dodecylo-1,2,4-triazole? What is the boiling point of 1-tetradecylo-1,2,3-triazole?

  3. At what temperature did the authors evaporates the solvent of CTA membranes. Is it possible to include the membrane morphology from SEM?

  4. In section 2.3, the authors used fix volume ratio for CTA, plasticizer and carrier. However, table 2 shows in wt%, what affects the transport of ions most for a different carrier in this work, carbon chain length and viscosity or the content of carrier in the membrane?

  5. In figure 2, the dimensions of the module can include in this diagram. How the authors put the CTA membrane in the module, please mention in section 2.4?

  6. In table 4, please include the fix parameters in the testing on the table footnote.

  7. Figure 3 is suggested to mention first before figure 4. Why 2cm^3/min is the optimum flow-rate? The mechanism is not clear in section 3.1.

  8. The authors found out that shorter the carbon chain of triazole is better because of its low viscosity. If triazole with the number of carbon chain below 8, the transport of copper in might be better? Please give insight on this in section 3.4.

  9. In section 3.5, why the author chose to use TRIA-10?

Reviewer 3 Report

This paper deals with copper transport through PIM including triazole carrier. The experiment itself is carried out with careful treatments, but I think the approach is quite common, therefore not so much valuable information will be included in the manuscript. I have found a few important issues that will improve the manuscript.

  1. In introduction, authors states the application of PIM after leaching. In this experiment, authors do not use the model leachate as feed phase.
  2. In Experimental, authors used the spiral membrane. There are no descriptions how to prepare the spiral membrane.
  3.  Line 221. "Each sample was analyzed three times." Which does this sentence means that "same sample was analyzed three times" or "three different samples under same condition were analyzed"?
  4. Figures 3, 6 and 9 show the copper concentration in the receiving phase. I think copper concentration in the feed phase should be also shown because of the material balance.
  5. Control experiment should be shown in Fig. 3 to calrify facilitated transport.
  6. Unlike previous PIM transport experiments, the permeation of copper stops at the equal concentration in both phases because of no counter transport. I think this point is practically much demerit.
  7. In Fig.10, decrease in flux with increasing alkyl chain length is caused by the increase in viscosity. However, in the temperature effect, authors claimed the chemical reaction control the flux. Authors discuss theoritically the contribution of diffusion and recation to the flux.
  8.  line 592-: Authors results should be compared with those of other PIM.

Round 2

Reviewer 3 Report

Revised manuscript is accepted for publication.

Author Response

Thank you for accepting our explanations. The text has been additionally corrected from the linguistic side.